# Multi-Sensor-Based Hierarchical Detection and Tracking Method for Inland Waterway Ship Chimneys

Fumin Wu [1], Qianqian Chen [1,*], Yuanqiao Wen [2,3], Changshi Xiao [1,3,4] and Feier Zeng [1]

1   School of Navigation, Wuhan University of Technology, Wuhan 430063, China;
    wfm19970@whut.edu.cn (F.W.); cs_xiao@hotmail.com (C.X.); 304716@whut.edu.cn (F.Z.)
2   Intelligent Transportation Systems Research Center, Wuhan University of Technology, Wuhan 430063, China;
    yqwen@whut.edu.cn
3   National Engineering Research Center for Water Transport Safety, Wuhan 430063, China
4   Institute of Ocean Information Technology, Shandong Jiaotong University, Weihai 264200, China
*   Correspondence: chenqq@whut.edu.cn

**Abstract:** In the field of automatic detection of ship exhaust behavior, a deep learning-based multi-sensor hierarchical detection method for tracking inland river ship chimneys is proposed to locate the ship exhaust behavior detection area quickly and accurately. Firstly, the primary detection uses a target detector based on a convolutional neural network to extract the shipping area in the visible image, and the secondary detection applies the Ostu binarization algorithm and image morphology operation, based on the infrared image and the primary detection results to obtain the chimney target by combining the location and area features; further, the improved DeepSORT algorithm is applied to achieve the ship chimney tracking. The results show that the multi-sensor-based hierarchical detection and tracking method can achieve real-time detection and tracking of ship chimneys, and can provide technical reference for the automatic detection of ship exhaust behavior.

**Keywords:** ship exhaust behavior; detection and tracking; multi-sensor; deep learning; morphological operation

## 1. Introduction

The construction of the Yangtze River Economic Belt is one of the key strategies of the national cross-regional coordinated development, and both the "Yangtze River Protection" and the "Yangtze River Green Ecological Corridor" are the top priorities of the construction of the Yangtze River Economic Belt. The International Maritime Organization (IMO) has mandated a gradual reduction of nitrogen oxide and other types of gas emissions [1], and a regulation on sulfur emissions from ships sailing in global waters has been in effect since 1 January 2020 [2]. In addition, the design of ships' intake ports and the exhaust ports of the exhaust gas is being modified in accordance with the requirements of the International Maritime Organization (IMO) [3]. However, the detection of ship exhaust depends on high-sensitivity gas sensors, and it is difficult to obtain evidence. The Pankratova NV study showed that ship exhaust emission data are correlated with ship chimneys [4]; therefore, the method of tracking ship chimney detection based on computer technology is one of the most important tools for scientific and efficient regulation.

Ship chimney detection is the core research content of this paper, and ship detection is the prerequisite and a key technical point of ship chimney detection. Since the ship chimney has small target and inconspicuous features, and the known chimney dataset is very small, it is very difficult to detect the ship chimney directly; on the contrary, the ship has relatively large target and obvious features compared with the chimney, and the dataset is relatively large.

However, currently there are still difficulties and challenges in the field of computer vision for small target detection. In terms of visible images, both traditional manually

designed feature operator-based target detection and deep learning-based target detection methods have yet to improve the detection accuracy of small targets. In addition to the characteristics of small targets, the detection of inland river ship chimneys is also affected by the small feature information of ship chimneys. In terms of infrared images, the infrared camera has a small field of view, and the acquired image information is not rich. Although the infrared camera is more sensitive to the target in high temperature regions and the ship chimney is also a high temperature object, the simple use of an infrared camera to detect the chimney is less robust due to the high temperature of the ship itself or the ship's cargo exposed to the sun, as well as the influence of the background buildings and water reflections in the inland river.

Based on the above problems, we found that, on the visible band image, although the ship chimney target is small, the ship target is relatively large and rich in information, and the deep learning technique can be used to detect the ship on the visible band image first, with the aim of narrowing down the detection range of the ship chimney. Then, since the visible band is more sensitive to high temperature regions, the difficulty of detecting the chimney in a small area will be greatly reduced. Therefore, the detection of ship chimneys can eventually be achieved by combining the characteristics of different sensor images, thereby bringing convenience to the subsequent tracking.

The remainder of this paper is organized as follows. Some related works are introduced in Section 2. In Section 3, we will discuss the whole methodology of our algorithm. The experiment and model prediction performance is reported in Section 4. Finally, the work is concluded in Section 5.

## 2. Related Work

A large number of scholars have also conducted research on ship detection based on computer vision techniques. According to the type of technology used, this research can be divided into traditional-based methods and deep learning-based methods. Most of the traditional methods are designed to detect or recognize a specific scene. Arshad [5] et al. first processed the ship background image using morphological operations, and then used the Sobel operator to perform edge detection of the ship to discriminate it from its background, but it is not effective in the case of complex textures, which have more noise. Zhang X designed a rotated Gaussian mask to model the ship, and, at the same time, contextual information was used to enhance the perception of the ship [6]. Wang Y. [7] et al. proposed a ship detection algorithm based on a background difference method, but the algorithm was aimed at ship detection under a static background, and did not identify, classify, and track targets. Tang Y. [8] et al. adopted the fusion technology of multi-vision to analyze and detect ship targets by monitoring through local entropy and a connected domain, requiring two scans of images, which was inefficient, and the threshold had a great influence on the final effect. Shi W. et al. [9] proposed morphology with multiple structural elements to extract the edge features of ships by using different structural elements, which can fully retain various details of ships while filtering out background noises such as waves, but it is difficult to detect small targets.

In addition to the traditional vision technology-based methods mentioned above, deep learning technology-based methods are the mainstream ship detection methods at present. Excellent target detection methods based on deep learning are the R-CNN series, YOLO series, and SSD series. Cui ZY used a pyramidal structure to connect the convolutional block attention module (CBAM) closely with each feature map connected from top to bottom of the pyramidal network in order to extract rich features containing resolution and semantic information for multi-scale ship detection [10]. Subsequently, Cui ZY proposed a center net-based large SAR image ship detection method for locating the centroid of the target by key point estimation, which can effectively avoid the missed detection of small target ships [11]. Differently, Chen XQ used a convolutional neural network in the YOLO model to extract multi-scale ship features from the input ship images. Then, multiple bounding boxes (i.e., potential ship positions) were generated based on the target confidence, and, finally,

the background surround box interference was suppressed to obtain the ship positions in each ship image. Finally, Chen XQ analyzed the spatio–temporal behavior of ships in continuous ocean images based on the ship's kinematic information [12]. Shao ZF used the CNN framework based on depth features, saliency maps, and coastline prior. This work integrated ship discriminative features to detect ship class and location [13]. Yang X proposed a dense feature pyramid network to detect ships in different scenarios, including in the ocean and at ports, in order to solve the problem caused by narrow ship width [14].

In recent years, deep learning methods have been successfully applied to ship detection in synthetic aperture radar (SAR) images. Wei SJ proposed a high-resolution ship detection network based on high-resolution and low-resolution convolutional feature mapping for ship detection in high-resolution SAR images [15]. Similarly, Lin Z, et al. proposed a new fast R-CNN-based network structure based on high-resolution SAR images to further improve ship detection performance by using a squeeze excitation mechanism [16,17]. Jin L., et al. used the SSD model and added a feature fusion module to the shallow feature layer to optimize the feature extraction capability for small objects, and then added the squeeze and excitation network (SE) module to each feature layer to introduce an attention mechanism for the network to achieve small-scale ship detection in remote sensing images [18,19]. Wang Y combined single-shot multibox detector (SSD) with migration learning to solve the ship detection problem in complex environments, such as oceans and islands [20]. Sun J, based on the SSD model, integrated expansion convolution with a multiscale feature fusion to improve small target detection accuracy [21]. Not coincidentally, Chen P, to improve the small target detection accuracy, embedded the elemental pyramid model into the traditional RPN, and then mapped it to a new elemental space for object recognition [22]. The detection of multiscale SAR ships remains a great challenge due to the strong interference and wide variation of scales in the offshore background.

This paper proposes a multi-sensor hierarchical detection tracking algorithm based on deep learning to detect and track ship chimneys. Firstly, the first level detection uses a visible light image input deep-learning target detector to detect the ship target, so as to greatly reduce the target detection range and solve the problem of background interference. Then, in view of the problem that the chimney target is too small to be identified, infrared imaging is adopted for the second-level detection, with the first-level detection result used as the input of the second-level detection. The image is extracted through a two-step Ostu binarization algorithm, image corrosion, and expansion operation. Finally, according to the prior knowledge of the chimney orientation, the candidate area is bisecting to further reduce the detection range and extract the final chimney target, combined with area characteristics. The improved DeepSORT tracking algorithm is used to track the ship chimney, which provides some help for the ship exhaust monitoring.

## 3. Algorithm Design

### 3.1. Algorithmic Framework

The framework of the multi-sensor hierarchical detection and tracking algorithm is shown in Figure 1, which is divided into four parts, namely data input, detection, tracking, and data output. Among them, the input data are an infrared camera and a visible camera, and the detection stage is divided into primary detection and secondary detection. The primary detection uses the improved YOLOV3 which was proposed by Joseph Redmon in 2018 as the ship detector, which is improved from two aspects: the design of the a priori frame, and the output of the feature pyramid. The second level detection splits the ship area in the infrared camera according to the first level detection result, and then filters the background by Gaussian filtering and adaptive threshold selection algorithm to obtain the candidate area of the chimney, according to the a priori knowledge. It is known that the chimney detected in this paper is located above the ship area, so the area equalization method is used to narrow the detection range again. Finally, the maximum value of the contour area is calculated as the final detection result.

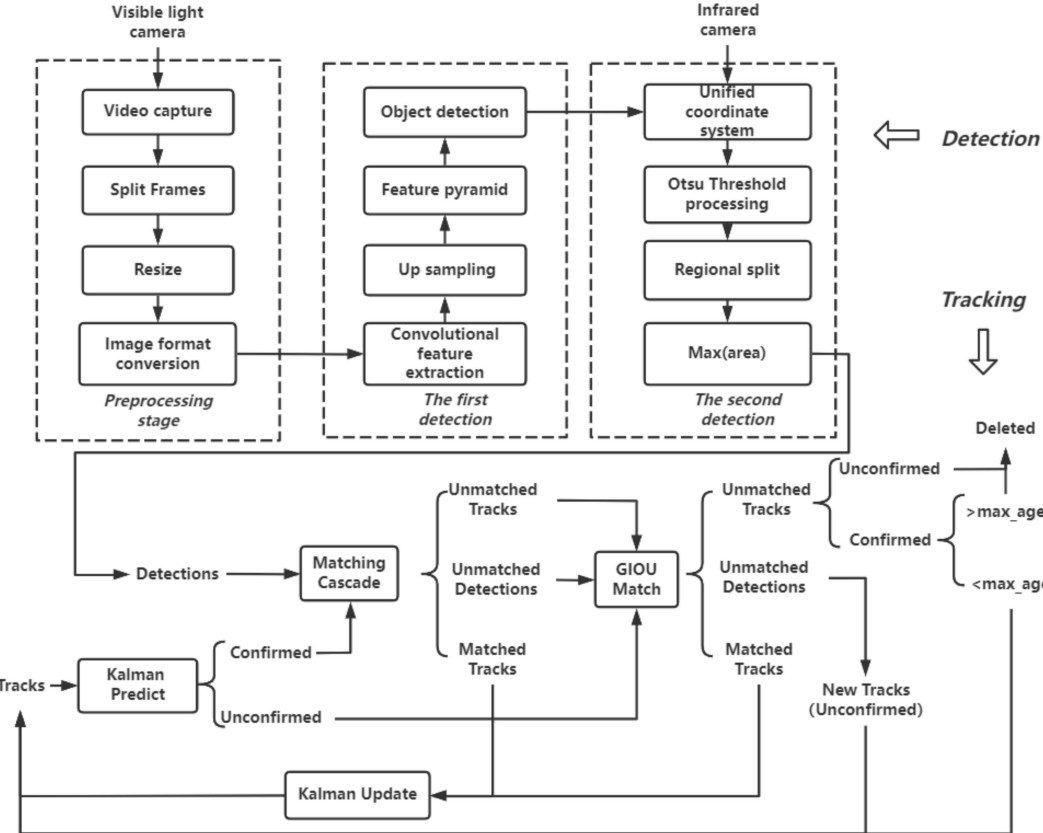

**Figure 1.** General framework of ship chimney detection and tracking algorithm.

The tracking is performed using the improved DeepSORT [23] algorithm, which mainly consists of a target detection module and a data association module. DeepSORT is used in the real-time target tracking process to first extract the depth features of the target, and then uses Kalman filtering to make predictions, correlate the sequence data, and perform the target matching. Mainly from the calculation of the cost matrix and association, the algorithm is improved. The main steps of the improved DeepSORT tracking are as follows:

(1) Create Tracks according to the results detected in the first frame, and initialize the Kalman filter. Tracks are initially in Unconfirmed state and can be converted to Confirmed state only if they are tracked successfully three times in a row.

(2) Calculate the cost matrix between the tracked target in Tracks and the detected target in the current frame using the improved GIOU.

(3) The cost matrix in Step (2) is input to the improved data association algorithm KM, and three kinds of matching results are obtained: the first category Matched Tracks is the traces matched to the detection results, indicating that the current frame tracks the target in the previous frame, and the values in Tracks are subsequently updated according to Kalman filtering. The second type, Unmatched Detections, is the detection result of unmatched tracks, which means that the target detected in the current frame is a newly appeared target, which is not related to the previous detection result, so a new tracking track needs to be added. The third type, Unmatched Tracks, is the trajectory with unmatched detections, which means that the trajectory existing in the previous frame is lost in the current frame, and if it is an Unconfirmed stable state, the trajectory is deleted directly. If it is a Confirmed stable state, the number of followed traces max_age is increased by 1. When the number of followed traces reaches 30 times, the Confirmed state is converted to an Unconfirmed non-stable state.

(4) For the Confirmed state, Tracks and Detections will use cascade matching to calculate the cost matrix. Cascade matching uses the appearance feature vector to calculate

the cosine similarity, and uses the Marxist distance to exclude the targets between frames that are far away from each other, where the appearance feature vector saves the feature vector of this target in the first 100 frames by default.

(5) There are also three types of cascade matching results: for the Unmatched Tracks and Unmatched Detections states, the algorithm re-calculates these two states together with the Unconfirmed state in Tracks using the GIOU association algorithm. For Matched Tracks states, the variable information in Tracks is updated by Kalman.

(6) The cost matrix in (5) is input into the KM algorithm, and the processing result is similar to step (3).

(7) Loop (4) to (6) steps until the end of the video frame.

*3.2. Improved YOLOv3-Based Ship Detection Network*

3.2.1. Anchor Improvements

A large number of experiments have shown that the selection and design of Anchor has had a large impact on the results of detection. Through the analysis of our own ship dataset, we know that the ship targets are larger, and the ship lengths and widths are more similar with horizontal orientation. By comparing the characteristics of the COCO dataset, we can see that the default Anchor of YOLOV3 does not meet our actual needs. Based on the above characteristics of the actual ship dataset, we made a specific design for the Anchor of the ship target, aiming to improve the speed and accuracy of ship detection.

In YOLO detection algorithm, the input image is divided into S × S grids, and each grid is called a Grid Cell. Each Grid Cell is responsible for detecting a target on which the center of the object falls. Each Grid Cell has a prediction box, which we call Anchor, and the number of Anchor for each Grid Cell is different in different versions. In YOLOV1, the image is divided into 7 × 7 size, and each grid is fixed with only two Anchors with different aspect ratios. Each Grid Cell can predict only one category, so the detection accuracy is low in scenes with dense targets. In YOLOV2, the authors used clustering to cluster the real target aspect ratios of the dataset into five classes by default, thus introducing five Anchors for each Grid Cell, and improving the detection capability for dense objects. In YOLOV3, the authors reduce the number of Anchors for each Grid Cell to three different scales, and introduce the concept of multi-scale feature map fusion to detect targets at different scales with three different scales, so the number of Anchor for each Grid Cell increases to nine.

The targets detected in this paper are ships, which generally have an aspect ratio greater than 1, i.e., the detection frame rectangle is longer than wide, as shown in Figure 2, where the upper left corner shows the distribution of the number of ship types, the upper right corner shows the distribution of the rectangular frame of the ship training set, and the lower left corner shows the distribution of the target center x and y, where the horizontal and vertical coordinates are the ratio of x and y to the actual width and height of the image. The same is true for the lower right corner, where the original width of the image is 1920 pixels and the height is 1080 pixels. From the statistical results, we can see that most of the ship widths are distributed around 0.1~0.3, i.e., 192~576 pixels wide, and the heights are distributed around 0.02~0.1, i.e., 22~108 pixels high. In order to make our designed Anchor aspect ratio closer to the actual ship detection application, the number of each Grid Cell was reduced from three to two, and we kept the default Feature Map with two different scales due to the "small and large" characteristics in inland waters. As a result, the number of Anchors was reduced from the default nine to four. In order to make our designed Anchor aspect ratio closer to the actual ship detection application, we first clustered the aspect of the Bounding Box of the dataset, where there are multiple clustering methods. We borrowed the idea from the YOLOV2 authors, and used k-means algorithm to cluster the data into two classes, and obtained the original dimensions of Anchor for each Grid Cell as (384, 54) and (1152, 216).

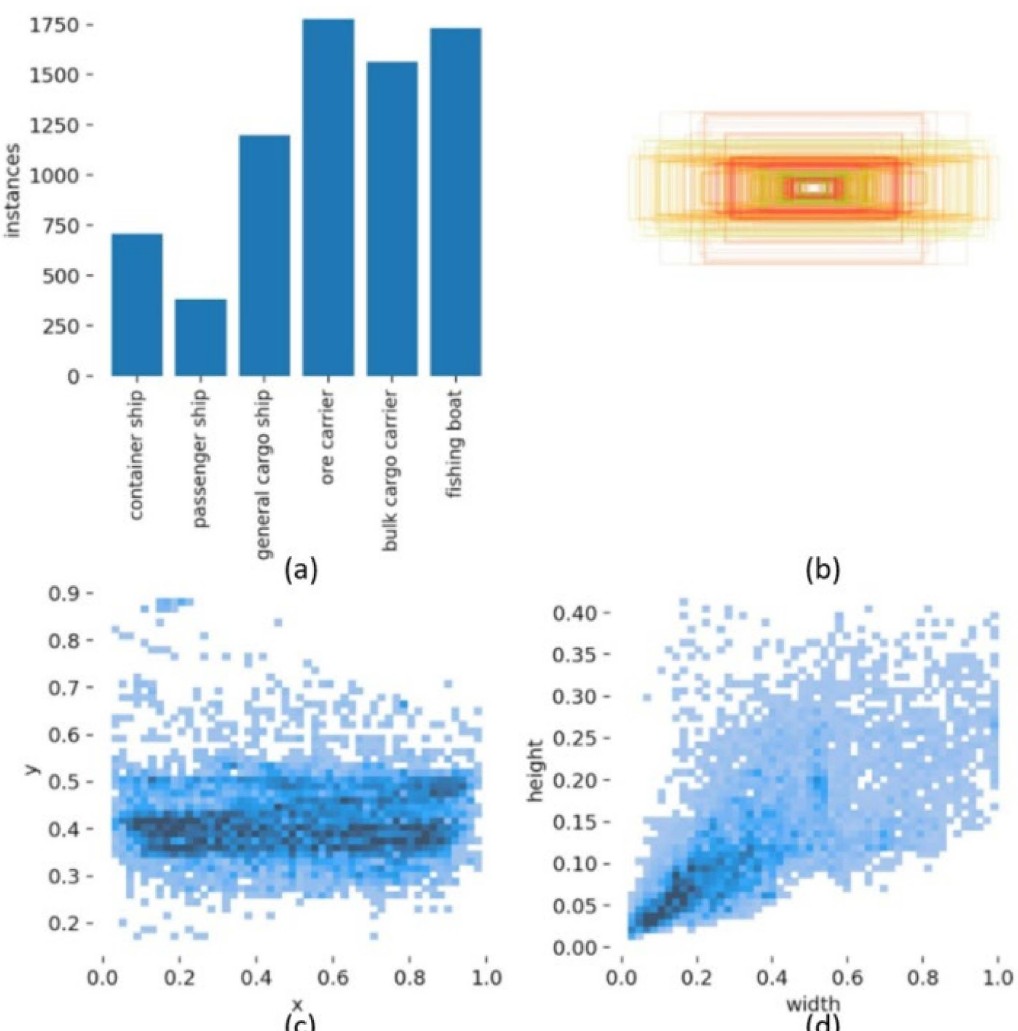

**Figure 2.** Distribution of Anchor information of a ship dataset. (**a**) Number of ships by class. (**b**) Statistics of Anchor shape. (**c**) Statistics of anchor X y center coordinates (**d**) Statistics of Anchor width height.

### 3.2.2. Improvement of Feature Pyramids

In YOLOV3, in order to make the detection of objects of different sizes, after the feature extraction network, the features of different feature extraction layers were fused to form new feature maps through Concat and upsampling operations. These different feature maps have the same depth, but different sizes. Fresh feature maps of different sizes, as well as the network structure in YOLOV3, are shown in Figure 3.

The light yellow part of the figure is for the three different scales of $13 \times 13$, $26 \times 26$, and $52 \times 52$. In these different scales, the size of each Cell is inversely proportional to the size of the scale, and for the large scale of $52 \times 52$, the corresponding size of each Cell is small, while for the small scale of $13 \times 13$, the size of each Cell is large. The small-scale Cell contains less information, and is therefore more suitable for detecting small objects, while the large-scale Cell incorporates more information, and is therefore more suitable for detecting larger objects, as shown in Figure 4. For the large ship in the bottom corner, a $13 \times 13$ feature map is generally available, while for the small target ship in the middle, a $26 \times 26$ feature map is generally available.

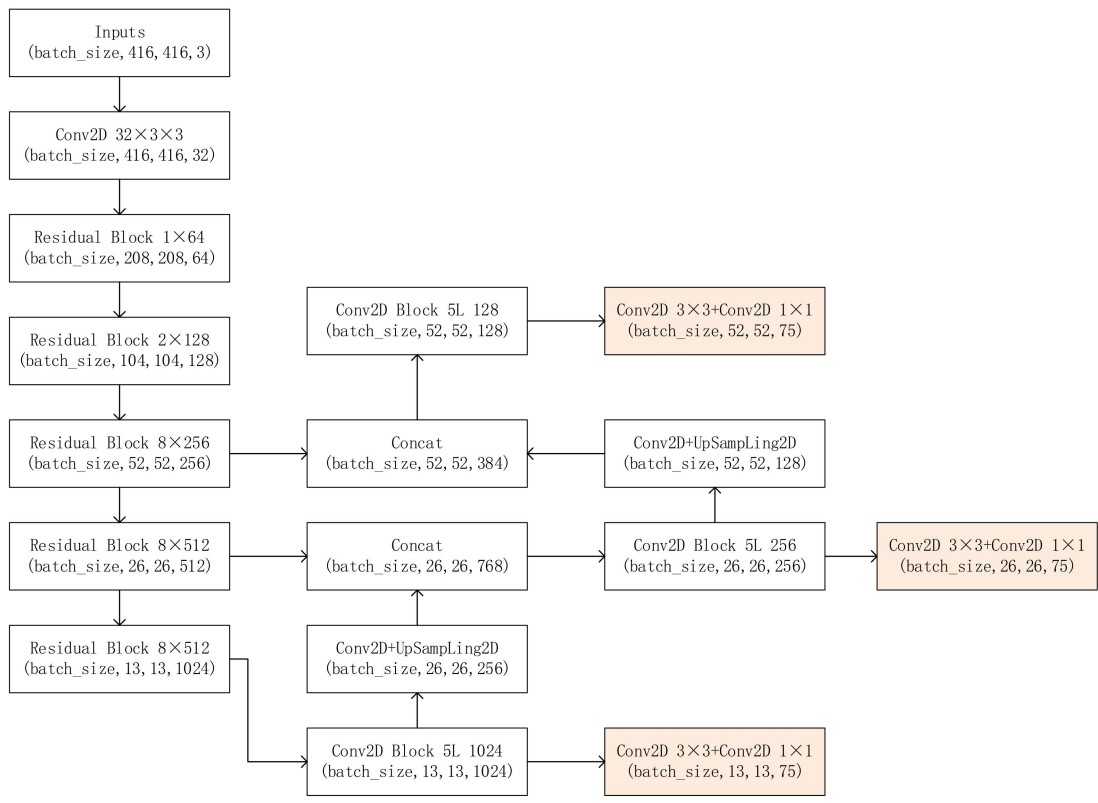

**Figure 3.** The network structure of YOLOV3.

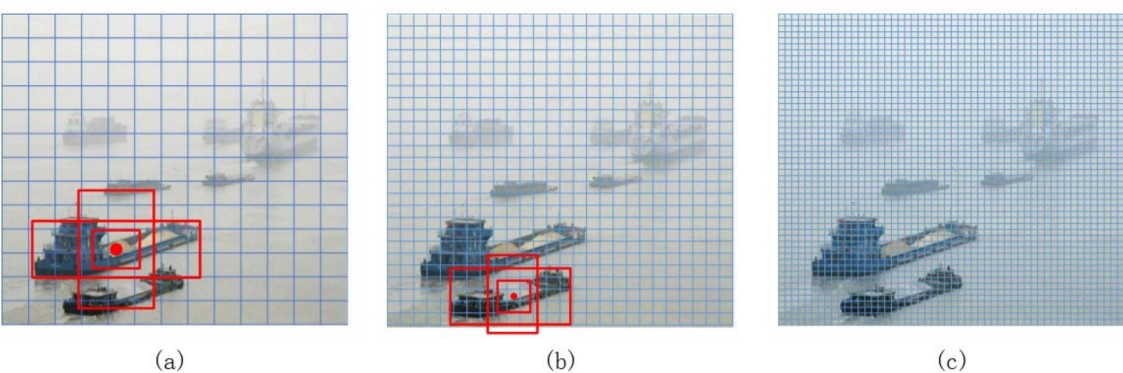

**Figure 4.** Output diagram of different sizes. (**a**) 13 × 13 grid cell. (**b**) 26 × 26 grid cell. (**c**) 52 × 52 grid cell.

Through analyzing the self-collected ship dataset in this paper, we can see that, in terms of species, the species of ships is much smaller than the open source generic dataset; in terms of scenarios, the river channel in inland waters is limited, and ships can only travel in the area. With the shore camera as the reference point, the width of the river channel greatly limits the size variation of ships, and most of the ships in inland waters are larger in size and belong to large targets, so we can delete the 52 × 52 feature maps used to detect small targets. Just keep the 13 × 13 and 26 × 26 feature maps. This optimization can reduce the parameters for network training, as well as speed up the training of the network. In addition, since the number of feature maps is reduced from three to two, the number of Anchor corresponding to each feature map is also reduced from three to two, so the original 3 × 3 = 9 frames to be detected is reduced to 2 × 2 = 4 frames to be detected when calculating the detection frames. This will greatly reduce the amount of calculation, as well as improve the detection speed of the ship. To sum up, the complete network structure after the improvement designed in this paper is shown in Figure 5.

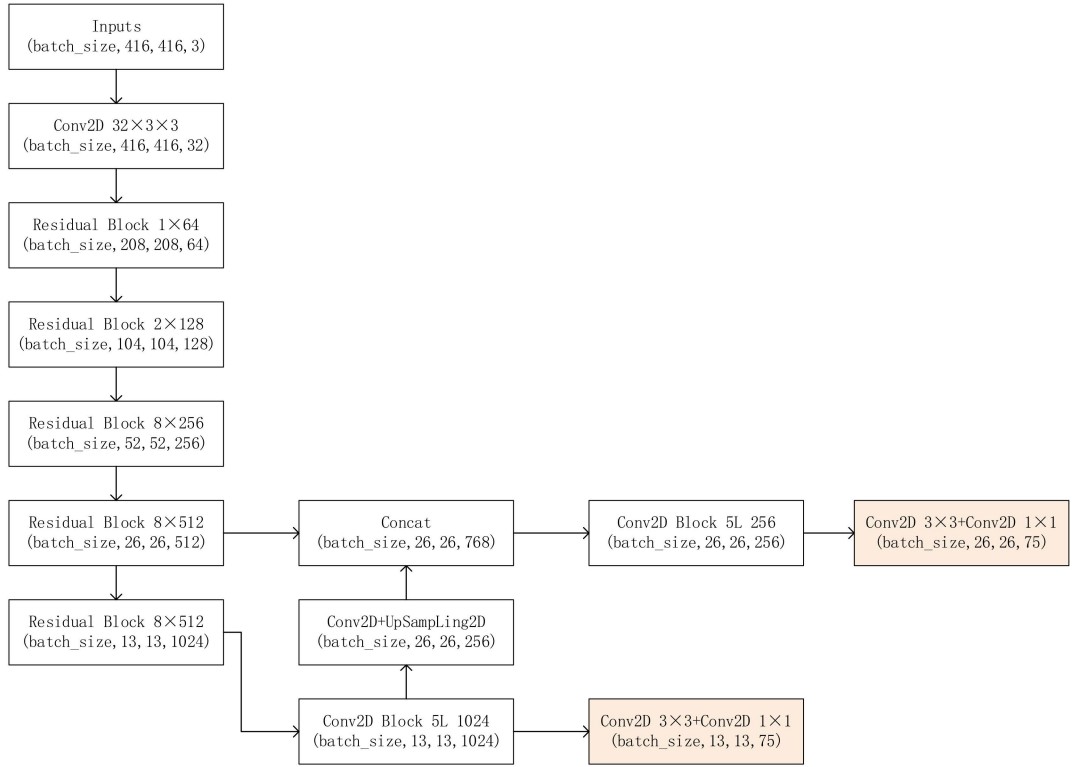

**Figure 5.** Improved YOLOv3 network.

As shown in Figure 5, the input size of the image is 416 × 416 pixels, and after five down-sampling calculations, a feature layer of size 13 × 13 is obtained, which detects large targets. Ship targets are relatively large targets, so a 52 × 52 feature layer will increase the number of parameters of the model and reduce the detection speed. Therefore, only two output layers of 13 × 13 and 26 × 26 are considered for retention. The goal of reducing the number of parameters and operations is achieved by reducing the number of feature layers to improve the network detection speed.

### 3.3. Chimney Detection with Fused Infrared Images

3.3.1. Threshold Processing

The video saved by the infrared heat-sensing camera used in this paper was later processed and saved locally as an RGB three-channel image as well. In order to facilitate the subsequent thresholding, the RGB image needed to be converted to a grayscale image. The conversion of RGB to gray scale image is represented by Equation (1)

$$\text{Gray} = \text{R} \times 0.299 + \text{G} \times 0.578 + \text{B} \times 0.114 \tag{1}$$

After the grayscale processing, a bimodal image can be obtained by counting the individual grayscale values, and due to the processing of bimodal images, this subsection uses Otsu's algorithm, which attempts to find a threshold that minimizes the weighted intra-class variance given by the relation:

$$\sigma^2 = \omega_1 \cdot (\mu_1 - \mu_0)^2 + \omega_2 \cdot (\mu_2 - \mu_0)^2 \tag{2}$$

where $\sigma^2$ is the interclass variance of foreground and background, $\omega_1$ and $\omega_2$ represent the proportion of background and foreground pixels in the total image, $\mu_1$ and $\mu_2$ represent

the average grayscale of background and foreground, respectively, and $\mu_0$ represents the average grayscale of the whole image. Expanding Equation (2) yields:

$$\begin{aligned}\sigma^2 &= \omega_1 \cdot \mu_1^2 + \omega_2 \cdot \mu_2^2 \\ &\quad -2(\omega_1 \cdot \mu_1 + \omega_2 \cdot \mu_2) \cdot \mu_0 + \mu_0^2\end{aligned} \tag{3}$$

According to the mathematical definition formula of expectation $E(X) = \sum\limits_{k=1}^{\infty} x_k \cdot p_k$, we can deduce that:

$$\mu_0 = \omega_1 \cdot \mu_1 + \omega_2 \cdot \mu_2 \tag{4}$$

Bringing (4) into (3), $\sigma^2 = \omega_1 \cdot \mu_1^2 + \omega_2 \cdot \mu_2^2 - \mu_0^2$ is again replaced using the relationship between Equation (4) and $\omega_2 = 1 - \omega_1$:

$$\begin{aligned}\sigma^2 &= \omega_1 \cdot \mu_1^2 + \frac{\omega_2^2 \cdot \mu_2^2}{1-\omega_1} - \mu_0^2 \\ &= \omega_1 \cdot \mu_1^2 + \frac{(\mu_0 - \omega_1 \cdot \mu_1)^2}{1-\omega_1} - \mu_0^2 \\ &= \frac{\omega_1}{(1-\omega_1)} \cdot (\mu_1 - \mu_0)^2\end{aligned} \tag{5}$$

Using Equation (5), we only need to count the pixels before the current iteration of grayscale, which greatly improves the efficiency of the program.

As can be seen in Figure 6, some background noise points can be effectively removed after Gaussian filtering. Compared with a fixed threshold, the Ostu algorithm is more likely to try to find a threshold to reasonably separate the foreground and background.

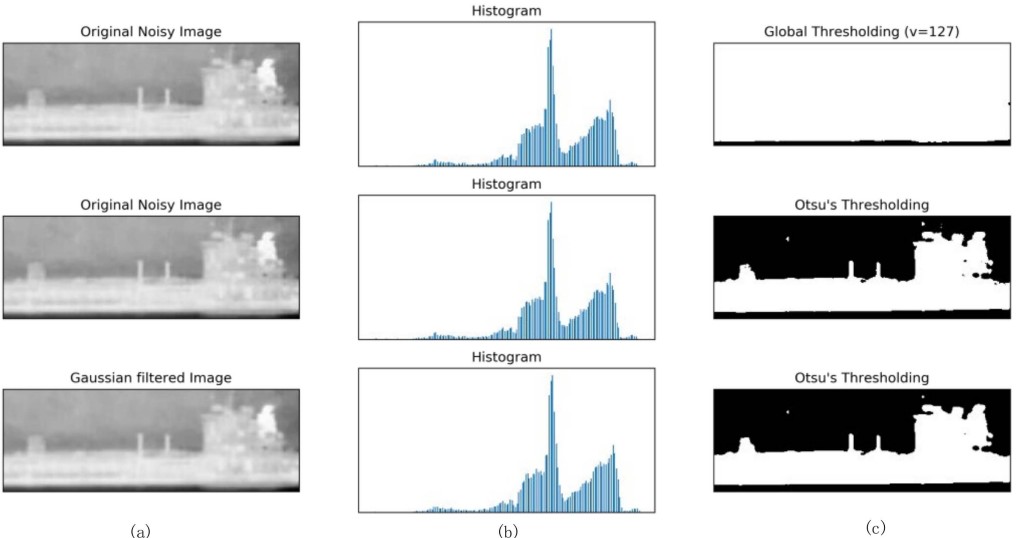

**Figure 6.** Comparison of infrared image binarization. (**a**) input images. (**b**) histogram. (**c**) Processing results of thresholds.

### 3.3.2. Coordinate Fusion

In order to collect experimental data, we have independently developed a set of experimental systems, which consists of a visible camera, an infrared camera and a gimbal that can be rotated coaxially, which can locate and track the target in real time. The visible camera has a resolution of $1920 \times 1080$, and the thermal imaging camera is a custom thermal imaging camera from Golder Infrared with a resolution of $640 \times 512$ resolution and a rotation angle of $-120$–$120°$ for the gimbal. The multi-sensor coaxial rotation system is shown in Figure 7.

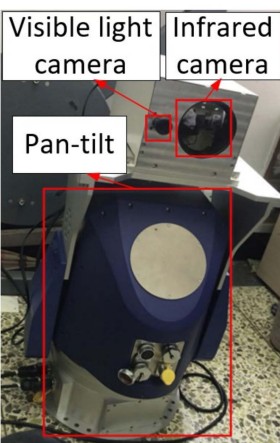

**Figure 7.** Multi-sensor coaxial rotation system.

Therefore, the coordinates of the same object in different cameras in the same frame are represented differently, as shown in Figure 8.

$$x_2 = \frac{x_1}{1920} \times 640 \tag{6}$$

$$y_2 = \frac{y_1}{1080} \times 512 \tag{7}$$

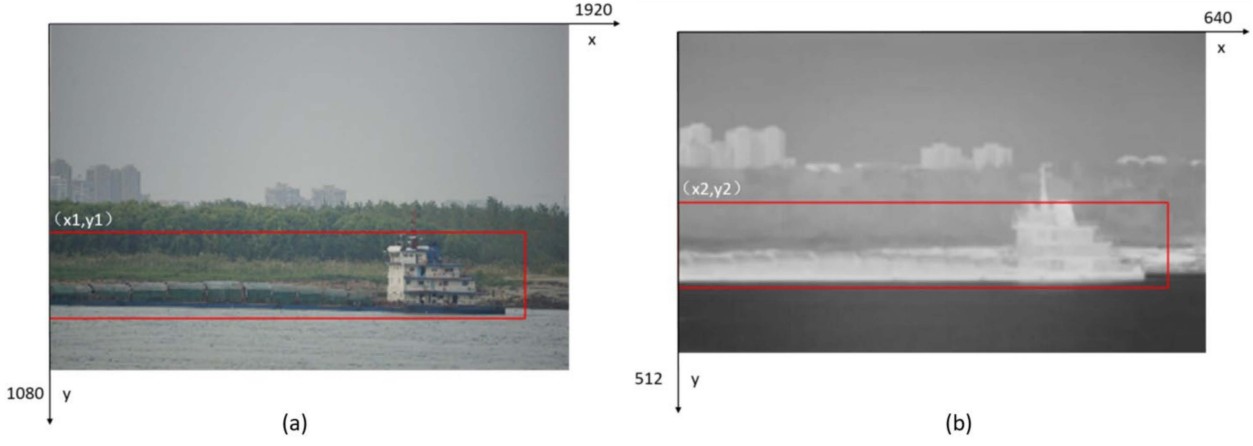

**Figure 8.** Schematic diagram of the same ship position in different coordinate systems. (**a**) Detection results in the visible light camera coordinate system. (**b**) Detection results in infrared camera coordinate system.

Therefore, this paper needs to convert the coordinates to ensure the accuracy of the search area of the ship's chimney. For different size images, the size is different, but the position of each coordinate point relative to the upper left corner (zero point) is the same after conversion to a right-angle coordinate system, so the coordinates can be converted using the scale relationship. For the image, the coordinate system is two-dimensional, so it needs to be converted separately in the $x$ and $y$ directions. Supposing that the coordinates of the same object P $(x_1, y_1)$ on the 1920 × 1080 resolution image and $(x_2, y_2)$, the specific value of $x, y$ of $(x_2, y_2)$ should be as shown in the operation of Equation (6), according to the image scale. By using the above-mentioned coordinate conversion equation after corresponding the visible image to the infrared band image, it aims to ensure that the location of the ship's chimney is found accurately, rather than deviations due to coordinate conversion.

### 3.4. Improving DeepSORT Algorithm

3.4.1. GIOU Loss Function

IOU (Intersection over Union), also known as intersection and merge ratio, is a measure of the accuracy of detecting the corresponding object in a given dataset. DeepSORT (Deep Simple Online and Realtime Tracking) uses the IOU of the detection frame and tracking frame as the loss matrix in the correlation algorithm. The input IOU ranges between [0, 1] with scale invariance, and the equation is shown in Equation (8):

$$\text{IOU} = \frac{S_A \cap S_B}{S_A \cup S_B} \tag{8}$$

where $S_A$ is the area of the predicted box, and $S_B$ is the area of the real box. If IOU is used as a measure of the overlap between boxes, the following problems will occur:

(1) IOU is always 0 when there is no overlap between the prediction box and the real box, as shown in Figure 9, state 1, where the red prediction box and the blue real box have no intersection, and the value of IOU is 0.

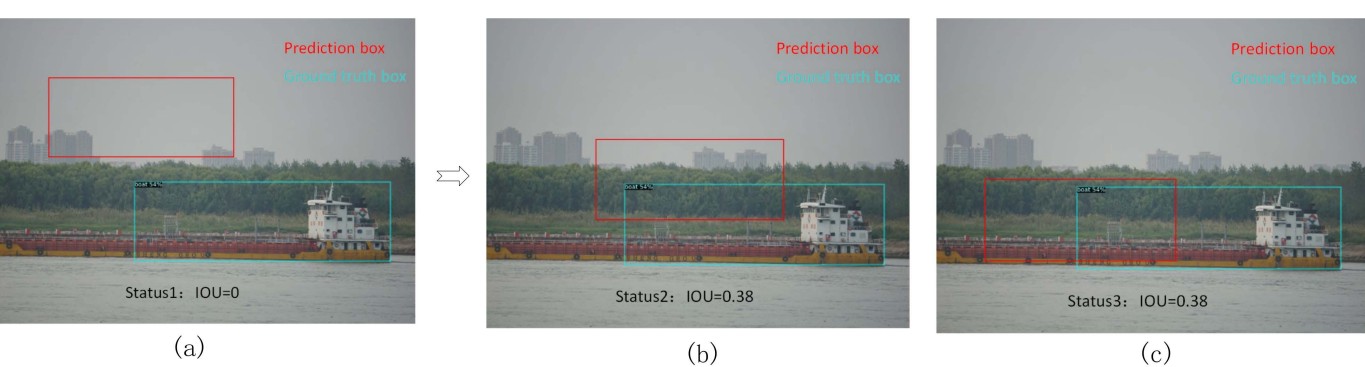

**Figure 9.** Schematic diagram of different overlapping shapes of IOU. (**a**) status1: IOU = 0. (**b**) status2: IOU = 0.38. (**c**) status3: IOU = 0.38.

(2) When the IOUs intersect and have the same value, it is impossible to distinguish the various cases of IOUs. There can be many kinds of overlapping shapes for the same IOU value, and they are different in effect. As shown in Figure 9 state 2 and state 3, the IOUs of both the prediction box and the real box in state 2 and state 3 are equal to 0.38, but state 2 is an up-and-down intersection, and state 3 is a horizontal intersection.

In order to solve the above problem, this paper uses GIOU (Generalized Intersection over Union) to replace IOU in the DeepSORT algorithm. GIOU loss focuses not only on overlapping regions, but also on non-overlapping regions, which distinguishes the cases with the same IOU but different forms of overlap, and solves the problem that there can be no gap between non overlapping frames. The value range of GIOU is [−1, 1] with the following formula.

$$\text{GIOU} = \text{IOU} - \frac{|C - (A \cup B)|}{|C|} \tag{9}$$

where $C$ is the smallest outer rectangle of the prediction frame and the target frame, as shown in the left of Figure 10. In Equation (8) is the difference set, as shown in the blue part in Figure 10.

As shown in Figure 11, suppose A is the ship target at frame n and B and C are the ship targets at frame n + 1, where the IOUs of A and B are $4/28 \approx 0.14$, and the IOUs of A and C are also $4/28 \approx 0.14$. Since their IOU values are equal, the difference cannot be measured if IOUs are used. However, the GIOU of A and B is $4/28 - (36 - 28)/36 \approx -0.08$, while the GIOU of A and C is $4/28 - (28 - 28)/28 \approx 0.14$, which shows that the correlation between A and C will be greater than that between A and B. Therefore, in the ship tracking task, we prefer to consider ship C as the target position of ship A in the next frame, which

is also consistent with the fact that inland ships travel slowly and have low deformation in the video sequence.

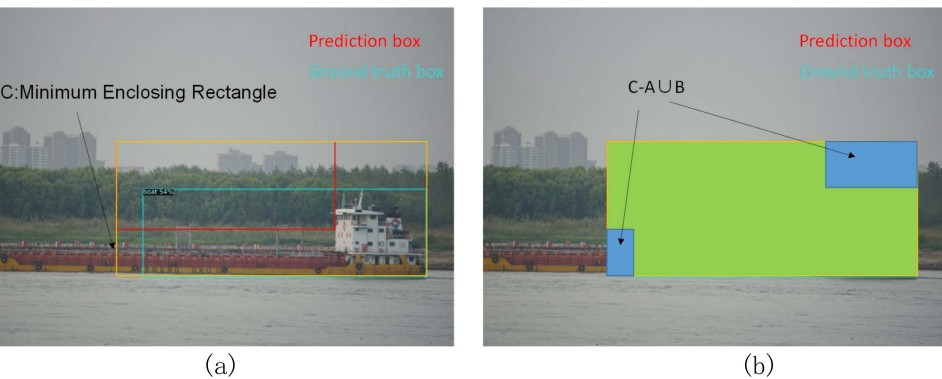

|  (a)  |  (b)  |

**Figure 10.** Schematic diagram of GIOU. (**a**) C is Minimum Enclosing Rectangle. (**b**) The areas of C-A∪B.

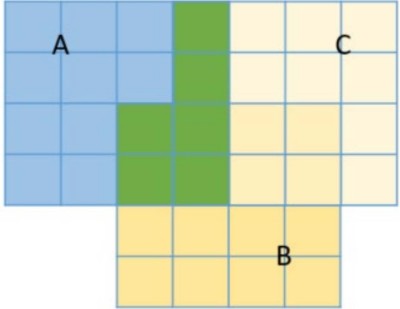

**Figure 11.** Schematic diagram of ship intersection between simulated frames. A is detection of n frame. B and C are detections of n + 1 frame.

### 3.4.2. KM Association Algorithm

In multi-target tracking tasks, the main purpose of data association is to perform matching of multiple targets between frames, including emerging targets, the disappearance of old targets, and the ID matching problem between the previous frame and the current frame. The DeepSORT default data association algorithm uses the Hungarian algorithm, and the core idea is to find the maximum matching algorithm of the augmented path for the bipartite graph. As shown in Table 1.

**Table 1.** Inter-frame target matching.

| n Frame | n + 1 Frame | | | |
| | N1 | N2 | N3 | N4 |
| GIOU | | | | |
|---|---|---|---|---|
| M1 | 0.8 | 0.6 | 0 | 0 |
| M2 | 0 | 0.6 | 0.9 | 0 |
| M3 | 0.9 | 0.8 | 0.5 | 0 |
| M4 | 0 | 0 | 0 | 0 |

M1~M4 are the four tracked targets in the nth frame, N1~N4 are the four newly detected targets in the nth + 1 frame, and the association degree index between the targets is measured by the GIOU loss function in the previous section. Since M4 is not associated with any detected targets in the new frames, M4 is the old target tracking loss case; N4 is the detected targets in the new frames, which belong to the new target emergence case and will be assigned new IDs to track. The association algorithm discussed in this section then

solves the matching problem between M1~M3 and N1~N3. If the Hungarian algorithm with no weight value is used, the matching results are generally: M1 matches with N1, M2 matches with N2, and M3 matches with N3 when the threshold value is taken as 0.5. The Hungarian algorithm considers both to be correlated as long as it is greater than the specified threshold, that is, it considers that M2 and N2 and N3 are matched while ignoring the fact that M2 is more correlated with N3. It is this matching method, which is regarded as leveling, that leads to low tracking accuracy.

The KM algorithm is an improvement of the Hungarian algorithm, in which the weights of the edge values are increased to achieve optimal weight matching based on the Hungarian algorithm. The steps to solve the target tracking problem involve using the KM algorithm [11]. The results detected in the *n*th and *n*th + 1 frames are used as vertices to form the point set M and the point set N, respectively, the GIOU of the detection frame and the prediction frame is used as the edge value connected between the two points, with the ID of each vertex in M set to $M_i$, the initial weight set to the maximum edge value W of the edge connected to that point, and each point in the point set N set to $N_i$, with the initial weight set 0. If the point set M is satisfied $M_i + N_i = W_{ij}$, then the $M_i$ and $N_i$ will be matched; if not satisfied, then the point set M in the conflict will be minus *d*, and the point set N in the conflict will be plus *d*, here set to 0.1. The specific process is shown in Figure 12.

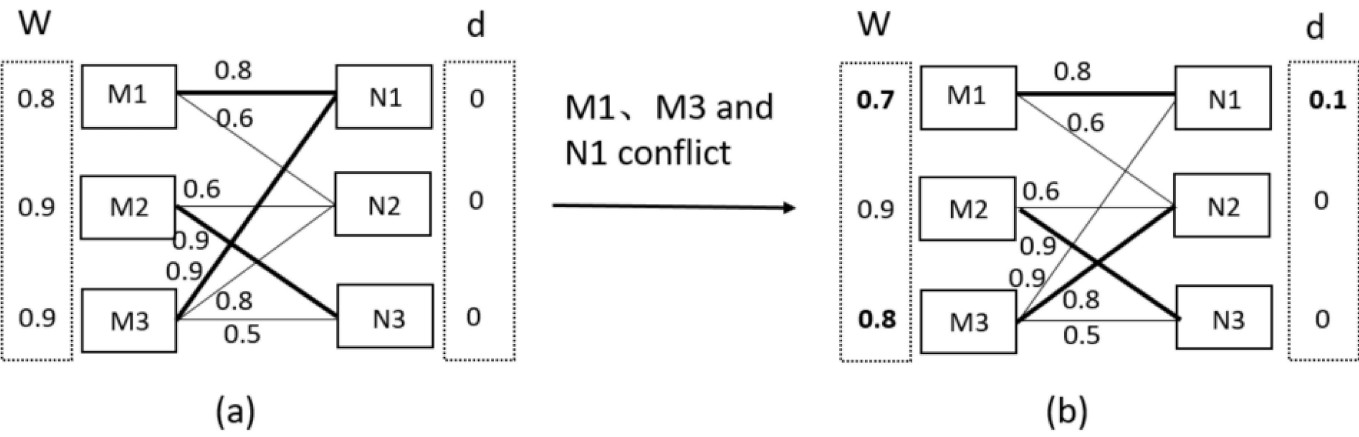

**Figure 12.** Schematic diagram of KM algorithm. (**a**) Initialize W, d. (**b**) Resolve conflict.

In Figure 11a, the KM algorithm assigns the initial value of W in the target M1~M3 from the maximum weight edge, and the initial value of d in the target N1~N3 is 0. After initialization, it is found that M1 and M3 are matched with N1, and try to change the edge weights of M1 and M3 to other values, but they do not satisfy $M_i + N_i = W_{ij}$. As such, a conflict arises. In order to resolve the conflict, the KM algorithm subtracts 0.1 from the W value of M1 and M3, and adds 0.1 to the d value of N1. At this point, M3 and N2 satisfy 0.8 + 0 = 0.8, and M1 and N1 also satisfy 0.7 + 0.1 = 0.8. The matching results obtained using the KM algorithm are: M1 matches N1, M2 matches N3, and M3 matches N2. The KM algorithm (total weight 0.8 + 0.9 + 0.8 = 2.5) is better than the Hungarian algorithm (total weight 0.8 + 0.6 + 0.5 = 1.9). 0.5 = 1.9) at matching yields with greater correlations.

## 4. Experimental Results and Analysis

### 4.1. Network Training Experiments

Due to the complex conditions of inland waters, the changeable weather, and the diversity of inland vessel types, datasets also require a large number of data sources. There are four main ways through which data sources were collected in this section: (1) ship images were collected and screened through search engines such as Baidu, Google, and Bing [24]; (2) high-definition surveillance cameras were built at fixed locations next to both banks of the Wuhan basin of the Yangtze River, which captured images cropped from the videos of ship navigation between 11 June 2019 and 17 November 2019; (3) the

image of the ship was captured with a digital camera in the Changjiang River Basin of Wuhan City, such as: erqi River Bank, Tianxingzhou Ferry Port, Hankou River Bank at a frequency of one per second, from 21 July 2020 to 26 November 2020. In this paper, data were collected from many locations and over a large time span, so the collected ship dataset meets the requirements of large data volume and sample types. The number of data sources is summarized as shown in Table 2. The ship types are divided into six categories as shown in Figure 13, and the statistical information of ship image data is shown in Table 3.

**Table 2.** Statistics of the number of data sources.

| Source | Quantity (Sheets) | Percentage (%) |
|---|---|---|
| Search Engine | 416 | 5.9 |
| Field shooting | 2061 | 29.4 |
| Surveillance video data | 4523 | 64.6 |

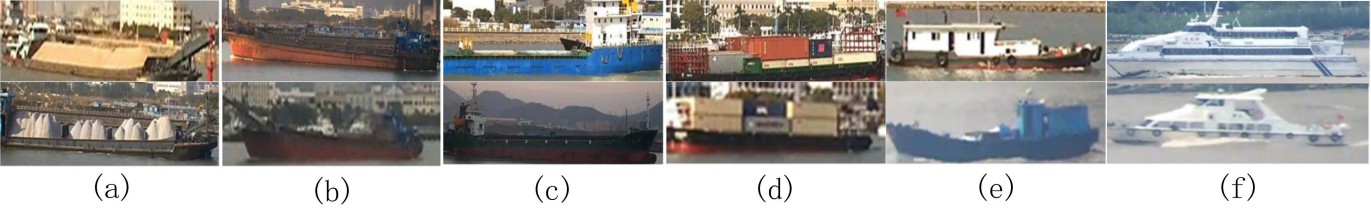

|     |     |     |     |     |     |
|-----|-----|-----|-----|-----|-----|
| (a) | (b) | (c) | (d) | (e) | (f) |

**Figure 13.** Vessel classification. (**a**) ore carrier. (**b**) bulk cargo carrier. (**c**) general cargo ship. (**d**) container ship. (**e**) fishing boat. (**f**) passenger ship.

**Table 3.** Number and proportion of ship types.

| Ship Type | Quantity (Sheets) | Percentage (%) |
|---|---|---|
| Ore Ships | 1751 | 25.0 |
| Bulk Carrier | 1498 | 21.4 |
| Miscellaneous Cargo Ships | 1149 | 16.4 |
| Container ship | 702 | 10.0 |
| Fishing boats | 1597 | 22.8 |
| Passenger Ship | 303 | 4.3 |

The ratio of the training set, validation set, and test set was 16:3:1. In order to ensure the objectivity of the experimental results, the hyper-parameter settings were consistent for different models, and some of the hyper-parameter settings related to the experiments are shown in Table 4.

**Table 4.** Hyper-parameter settings.

| Hyper-Parameter Name | Numerical Value |
|---|---|
| Batch Size | 64 |
| Weight Decay | 0.0005 |
| Momentum | 0.937 |
| IOU Threshold | 0.2 |
| Loss Gain | 21.35 |
| Epoch | 100 |
| Learning Rate | 0.002324 |

*4.2. Chimney Inspection Experiments*

The graded detection results are shown in Figure 14 below, where each column represents a set of data. The first and second rows show the raw data from the visible and infrared cameras, respectively; the third row shows the first-level detection, i.e., the result

of detection by the deep learning detector; the fourth row passes the first-level detection result and uses the two-step Ostu binarization algorithm to obtain the region with higher temperature, and further filters the non-chimney highlighted region by leveling the upper and lower regions; the fifth row filters the noise points through the image erosion operation, and expands the chimney candidate region through the expansion operation to expand the chimney candidate area; and the sixth row calculates the maximum value of the area of the chimney candidate area, and draws the contour of the maximum value as the final chimney detection area. After the first-level detection to narrow the range, the background interference can be reduced, and the accuracy of ship chimney detection can be improved.

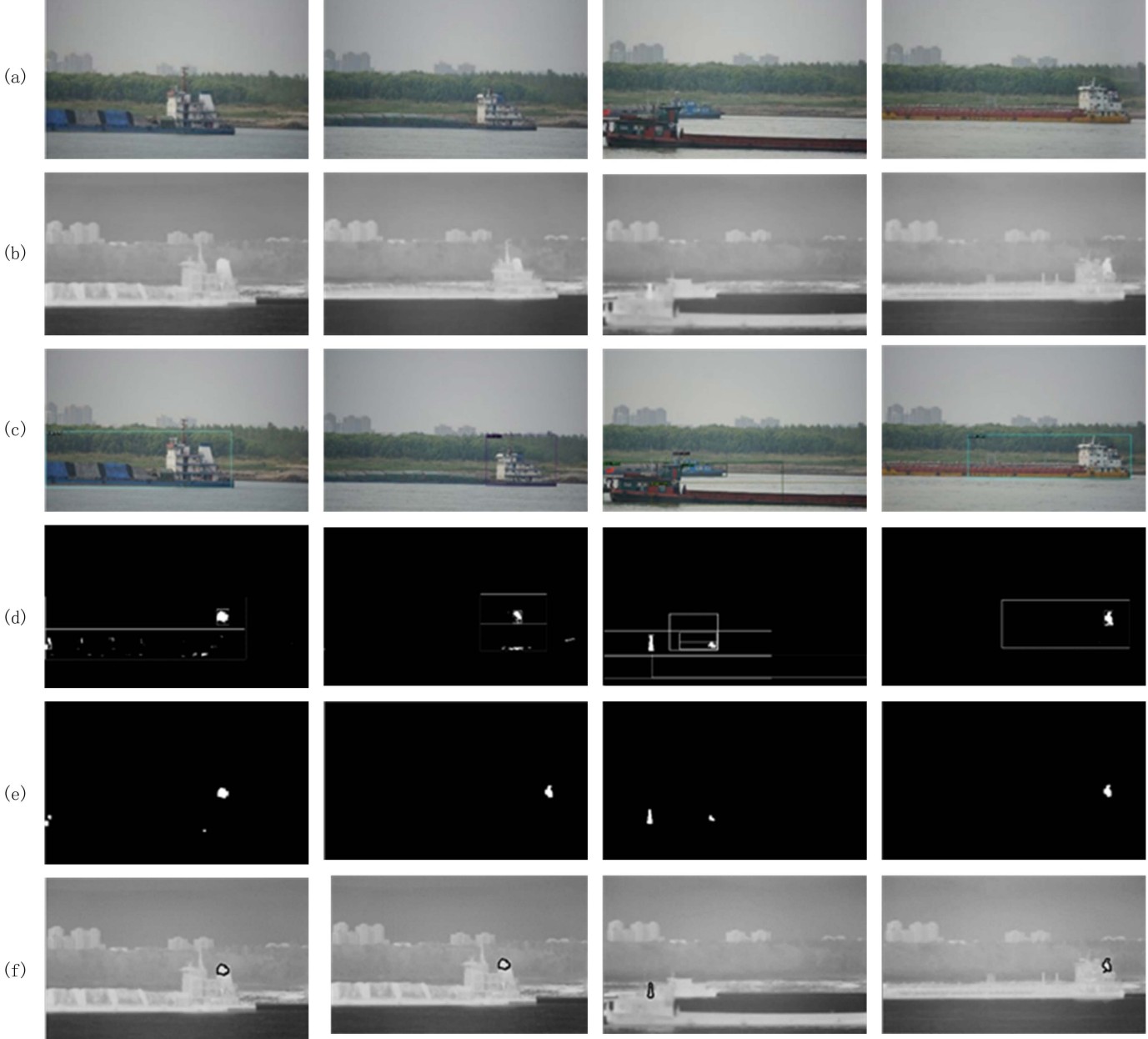

**Figure 14.** Chimney inspection. (**a**) Visible light input diagram. (**b**) Infrared camera input diagram. (**c**) First-level test results. (**d**) Optimization of detection range. (**e**) Secondary test results. (**f**) The results are displayed in the original image.

*4.3. Model Evaluation*

4.3.1. Evaluation Index of Ship Detection Model

We conducted numerical experiments on YOLOV3 [10] and our ship detection method. To evaluate the performance of the two models, we used the evaluation metrics, mainly Precision (accuracy), Recall (recall), mAP (mean average precision), and F1 (F1-Measure), and the calculation formula is as in Equation (9), where AP is the area value under the curve calculated by integration after the P-R curve is smoothed, and mAP is the mean value of AP for all categories.

$$
\begin{cases}
\text{precision} = \frac{TP}{TP + FP} \\
\text{recall} = \frac{TP}{TP + FN} \\
\text{F1} = \frac{2 \times \text{precision} \times \text{recall}}{\text{precision} + \text{recall}}
\end{cases}
\tag{10}
$$

where $TP$ means the sample is marked as positive, $FP$ means the sample is marked as positive by error, $TN$ means the sample is marked as negative by correct, and $FN$ means the sample is marked as negative by error.

After 100 rounds of iterative training using the migration study methodology, the training results are shown in Figure 15. The abscissa represents the number of training iterations, and the ordinate indicates accuracy, average recall, average accuracy and F1, respectively. As can be seen from the result graph, after the number of iterations reaches 40 rounds, the four basic parameters are stable at about 92%.

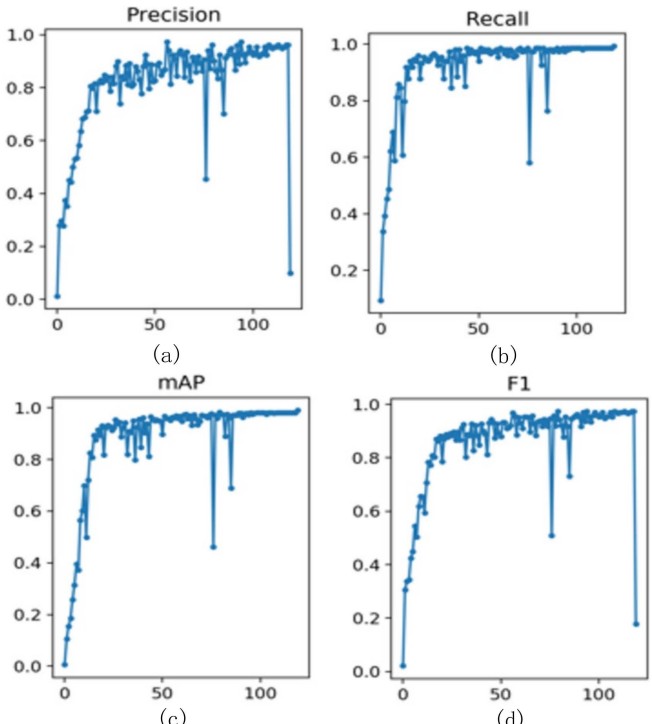

**Figure 15.** Results after 100 iterations. (**a**) Accuracy results. (**b**) Recall results. (**c**) mAP results. (**d**) F1 results.

In order to illustrate the effectiveness of our model, we carried out experiments in the same software and hardware environment, and the specific parameters are shown in Table 5. The calculation times of YOLOV3 model and our model were counted. To ensure the fairness of time cost, our calculation time was divided into two parts: training time, and verification time. The time consumed by each epoch is calculated in Table 4.

**Table 5.** Computational cost (SECOND).

| Method | YOLOV3 | | OURS | |
|---|---|---|---|---|
| Category | Train | Val | Train | Val |
| Time | 9.46 | 5.48 | 6.83 | 3.52 |

From the data in Table 4, we can see that the average calculation time of each epoch of YOLOV3 is 9.46, while the time of our model is lower, at 6.83. Compared with the calculation time of the validation model, our model is also faster. Therefore, our model has a good effect on real-time tasks, such as the detection and tracking of ship chimneys in inland rivers.

Under the same hyper-parameters, dataset, and the same experimental environment, the improved network and the original YOLOV3 network were compared experimentally, and the experimental results for each category are shown in Figure 16. The horizontal coordinates are the confidence values, and the vertical coordinates are the values of each metric at the current confidence level, and the overall values are shown in Table 6.

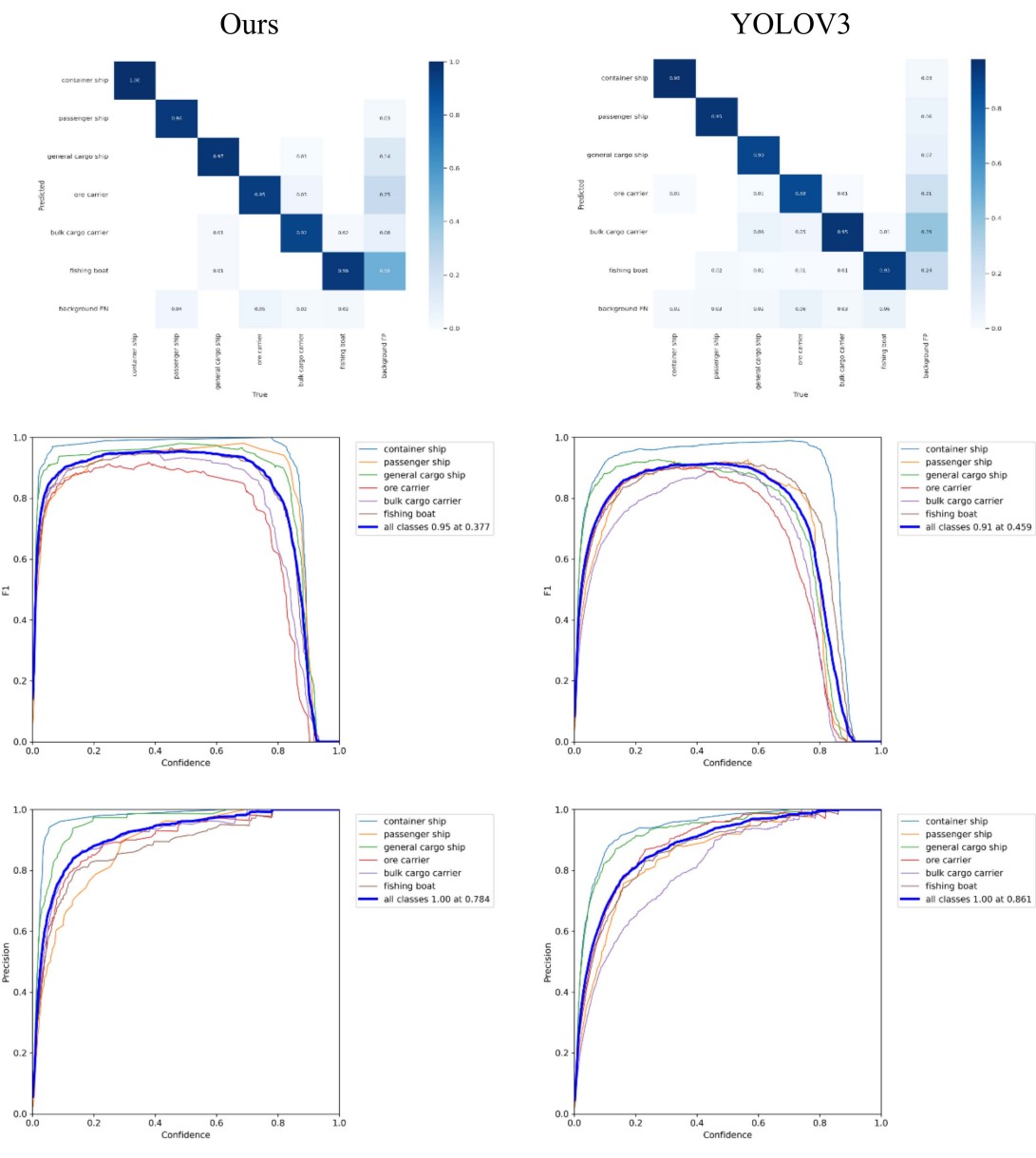

**Figure 16.** *Cont.*

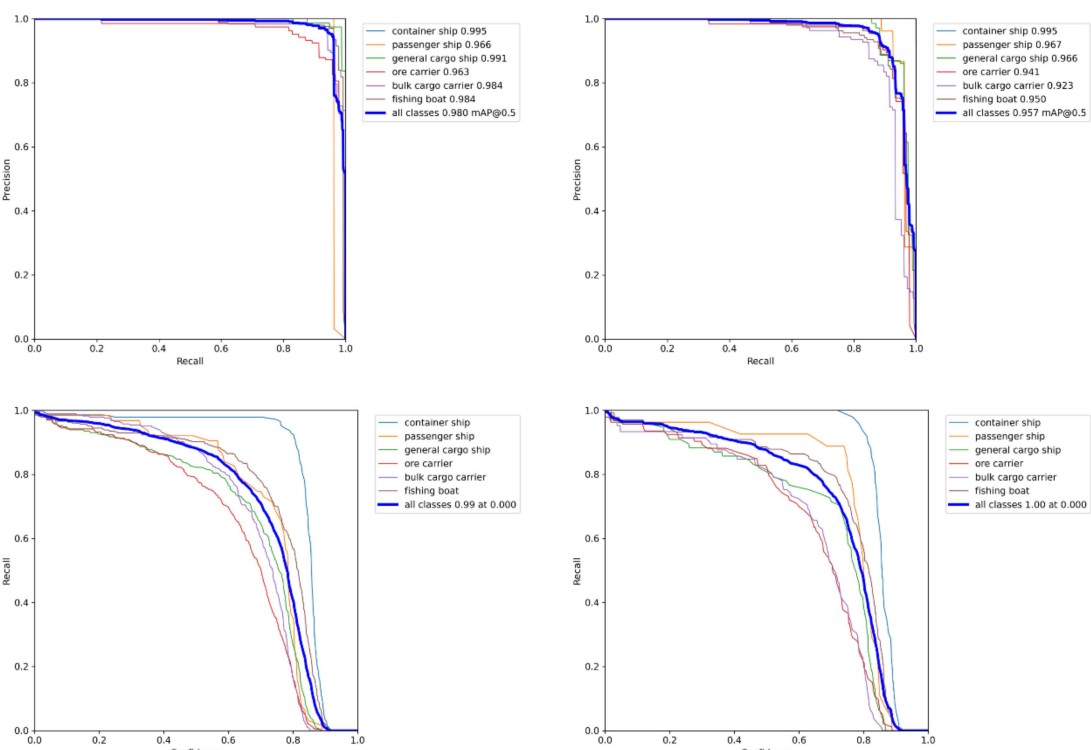

**Figure 16.** Graph of the results of various ship indicators on the test set.

**Table 6.** Overall index results on the test set.

| Evaluation Indicators | Ours | YOLOV3 |
|:---:|:---:|:---:|
| Precise | 0.95 | 0.93 |
| recall | 1.00 | 0.99 |
| mAP | 0.980 | 0.957 |
| F1 | 0.95 | 0.91 |
| FPS | 36 | 28 |

From the experimental results, our model can reach an accuracy of 1 when the confidence level is taken as 0.784, while the original model needs to be taken as 0.861 to reach 1. We can also see from the confusion matrix that the improved model has significantly less false detections than the original model, and the detector in this paper outperforms the YOLOV3 model in terms of accuracy, recall, mAP, and F1 indexes, especially in terms of detection speed, which is significantly higher than in the original model. The visualization of detection results is shown in Figure 17.

### 4.3.2. Ship Tracking Model Evaluation Index

In this paper, three metrics were chosen to evaluate the effectiveness of multiple object tracking: (1) ID switch indicates the number of times the target label is changed in a tracking track, and the smaller the value, the better; (2) multiple object tracking accuracy (MOTA) mainly considers the matching errors of all objects in the tracking process, mainly the FP, FN, and ID switch. MOTA gives a very intuitive measure of the performance of the tracking algorithm in detecting objects and maintaining the trajectory, independent of the progress of target detection. A larger MOTA value indicates a better performance of the model. MOTA is calculated as:

$$M_{OTA} = 1 - \frac{\sum (A_{FP} + A_{FN} + A_{ID})}{\sum A_{GT}} \tag{11}$$

where $A_{FP}$ is the number of false positive cases, $A_{FN}$ is the number of false negative cases, $M_{OTA}$ is the multi-target tracking accuracy, $A_{ID}$ is the number of ID switches, and $A_{GT}$ is the number of labeled targets; (3) FPS, the number of image frames per second processed by the model—the larger the value, the better the processing effect. To verify the performance of the method in this paper in chimney tracking, the test was conducted on the video surveillance data of the Yangtze River Bridge, and the test results are shown in Figure 18. The results are also shown in Table 5.

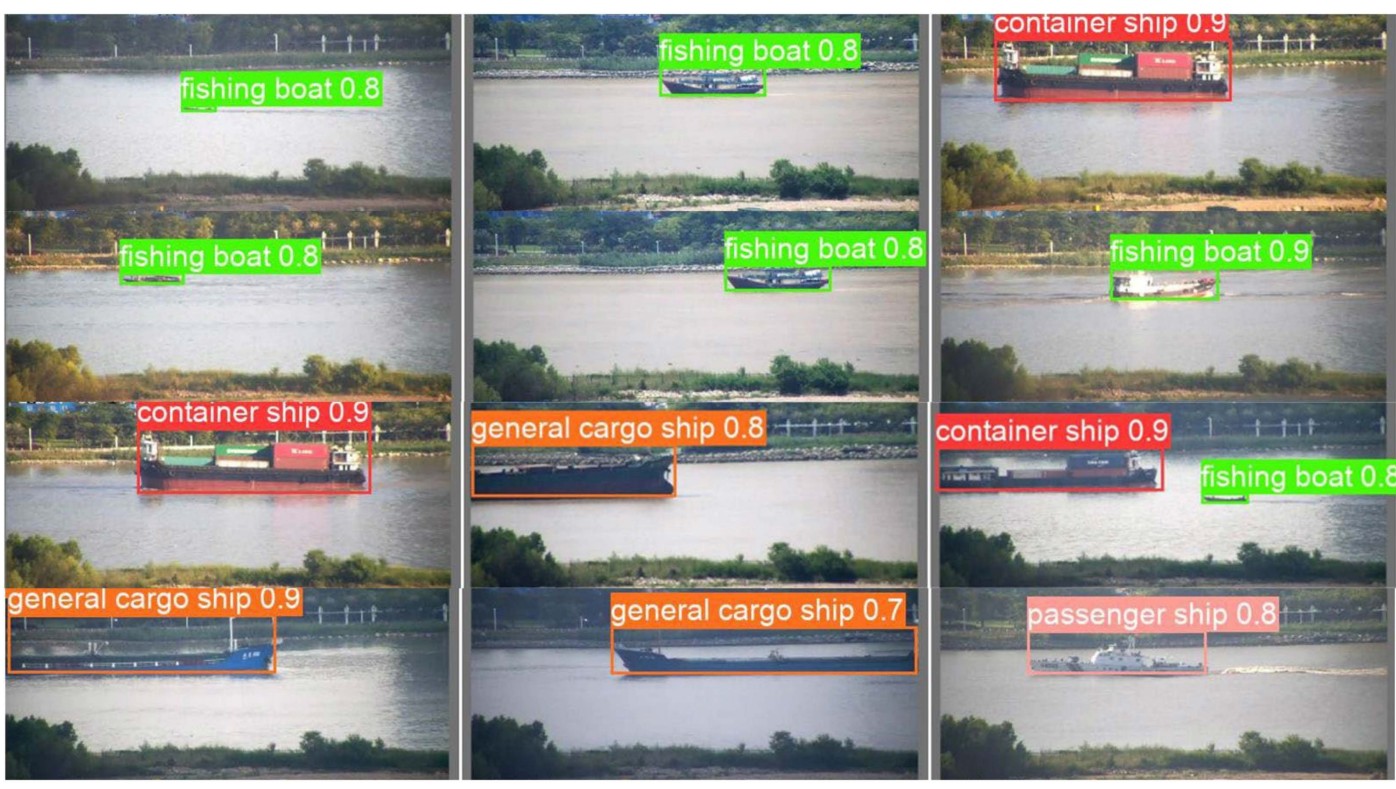

**Figure 17.** Visualization of test results.

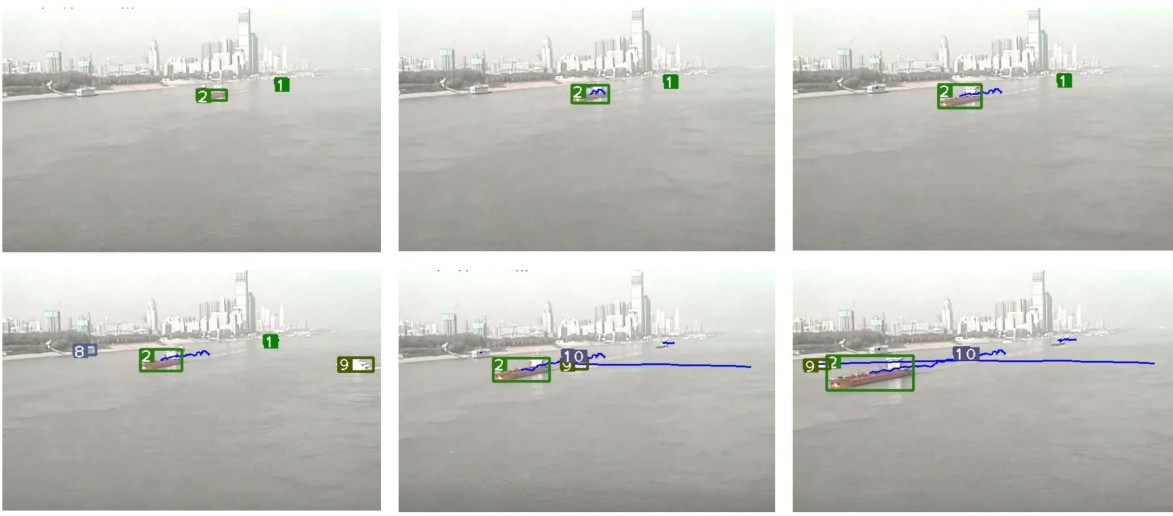

**Figure 18.** Vessel tracking visualization.

Among them, the blue box is the box detected by the deep learning detector, and the yellow box is the final box after the Kalman filter update. From the experimental results, it can be seen that the ID jump frequency decreases when the target is occluded, and the

accuracy rate increases by 0.04. The specific experimental results evaluation index is shown in the Table 7.

**Table 7.** Results of tracking metrics in the test set.

| Evaluation Indicators | Ours | DeepSORT |
| --- | --- | --- |
| Multi-target tracking accuracy | 0.75 | 0.71 |
| Number of target marker changes | 4 | 6 |
| Frame rate | 6.1 | 7.4 |

## 5. Conclusions

In this paper, we propose a deep learning-based multi-sensor hierarchical detection and tracking method for inland river ship chimneys, which makes full use of the image characteristics of different sensors, and combines the hierarchical idea to solve the problems encountered in practical engineering problems. The method uses visible images with rich feature information, combining deep neural networks to detect inland river ships, filtering irrelevant background information, and using the infrared camera's sensitivity to temperature to locate ship chimneys to ensure high accuracy of detection results under inland river waters with complex backgrounds. The reliability and practicality of the method are proved by field experiments. It makes a certain contribution to assisting the monitoring of automatic air pollution.

**Author Contributions:** Conceptualization, F.W. and Q.C.; methodology, F.W.; software, F.W.; validation, F.W.; formal analysis, Q.C.; investigation, F.W.; resources, Y.W and C.X.; data curation, F.W. and F.Z.; writing—original draft preparation, F.W. F.Z. and Q.C.; writing—review and editing, Q.C.; visualization, Y.W.; supervision, Y.W.; project administration, Y.W; funding acquisition, Y.W. and C.X. All authors have read and agreed to the published version of the manuscript.

**Funding:** This work was supported by the Natural Science Foundation of Shandong Province under Grant ZR2020KE029; by the National Natural Science Foundation of China under Grant 52001241; by the 111 Project (B21008); by the Zhejiang Key Research Program under Grant 2021C01010.

**Institutional Review Board Statement:** Not applicable.

**Informed Consent Statement:** Not applicable.

**Data Availability Statement:** The data presented in this study are openly available in [SeaShips] at [doi:10.1109/TMM.2018.2865686], reference number [24].

**Conflicts of Interest:** The authors declare that they have no known competing financial interests or personal relationships that could have appeared to influence the work reported in this paper.

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
