# Peer review of "Multi-Sensor-Based Hierarchical Detection and Tracking Method for Inland Waterway Ship Chimneys"

_jmse, doi:10.3390/jmse10060809_

Round 1

Reviewer 1 Report

This paper presents a model for the detection of ship chimnies based on multisensor data and deep learning algorithms. The paper is interesting but there are several improvements that should be done before publishing. 

The last paragraph of the Introduction describe the used methodology so it should be moved to the Methodology part. There are several papers published in this field. Please analyze them and include them in the discussion section.

The aim of this paper is not defined.

Please reference all algorithms that you have used in this paper (for example the YOLO).

 Include the accuracy assessment in Figure 1.

Did you train the algorithm for stretch or were you just fine-tuned? 

Please add implementation details.

The accuracy assessment should be included in the methodology, not in the result part. 

The biggest disadvantage of this paper is the lack of discussion. Please add a discussion section and compare your results with state-of-the-art papers in the field.

Author Response

  1. The last paragraph of the Introduction describe the used methodology so it should be moved to the Methodology part. There are several papers published in this field. Please analyze them and include them in the discussion section.

Answer: This is a very good suggestion. In the new manuscript, the description of the method has been moved to methodology, and the papers published in this field are discussed in the introduction. Details are described in Page 4 of the manuscript.

  1. The aim of this paper is not defined.

Answer: We appreciate the valuable comment by the reviewer. The purpose of this paper is to locate the inland waterway ship's chimney quickly and accurately, which is helpful to standardize the ship's emission behavior. The details about the fusion are described in Page 1 of the manuscript.

  1. Please reference all algorithms that you have used in this paper (for example the YOLO).

Answer: We appreciate the valuable comment by the reviewer. We have put forward references to the methods used in this paper. The details about the reference have been indicated in blue font.

  1. Include the accuracy assessment in Figure 1.

Answer: This is a very good suggestion. The new manuscript has made detailed modifications to Figure 1 and added accuracy evaluation. The details of  Figure 1 are described on Page 4 and Page 5 of the manuscript.

  1. Did you train the algorithm for stretch or were you just fine-tuned?

Answer: We appreciate the valuable comment by the reviewer. We just fine-tuned the algorithm. In future work, we train the algorithm for stretch and compare the detection effect of the two methods.

6. Please add implementation details.

Answer: We appreciate the valuable comment by the reviewer. The implementation of the algorithm is described in detail in Section 3.

7. The accuracy assessment should be included in the methodology, not in the result part.

Answer: We appreciate the valuable comment by the reviewer. We set evaluation criteria based on the accuracy and real-time performance of the predicted results, and we set up two model evaluation indexes for ship detection and ship tracking. The details of the evaluation criteria are described in Section 4.3 of the manuscript.

  1. The biggest disadvantage of this paper is the lack of discussion. Please add a discussion section and compare your results with state-of-the-art papers in the field.

Answer: We appreciate the valuable comment by the reviewer. The new manuscript adds the discussion part of the results. The details of the discussion are described in section 4.3 of the manuscript. And state-of-the-art papers in the field, the ship chimney detection method has not been studied, so it is difficult for us to find a comparable method.

Reviewer 2 Report

The work proposes a deep learning-based multi-sensor hierarchical detection method for tracking inland river ship chimneys is proposed to locate the ship exhaust behavior detection area quickly and accurately.

The topic fits the scope of the journal. 

The manuscript is well written, the structure of the paper is clear and the language is proper.

Besides the contributions are well delimited, the introduction section must be revised, rewrited and reorganised in order to clarify the motivation, objectives and the contributions in which advances the related work. It is not really clear the contributions compared to state of the art. Authors must describe and discuss the related scientific work in this field. 

I strongly suggest revisiting more references and related work in the topic covered by the paper.

The last paragraph of section 1 should write the organisation and structure of the manuscript.

All figures need to be improved to better quality.

The manuscript needs a revision in order to correct typos.

Author Response

  1. Besides the contributions are well delimited, the introduction section must be revised, rewrited and reorganised in order to clarify the motivation, objectives and the contributions in which advances the related work. It is not really clear the contributions compared to state of the art. Authors must describe and discuss the related scientific work in this field.

Answer: We appreciate that the reviewer pointed this out. The manuscript has been proofread and improved carefully, including the motivation, objectives, and contributions which advance the related work.

  1. I strongly suggest revisiting more references and related work on the topic covered by the paper.

Answer: We appreciate that the reviewer pointed this out. Relevant work parts have been added to the manuscript.

  1. The last paragraph of section 1 should write the organization and structure of the manuscript.

Answer: The organization and structure of the manuscript are added in the last paragraph of the first section. The details about the organization and structure are described in Page 5 of the manuscript.

  1. All figures need to be improved to better quality.

Answer: Thanks. The manuscript has been proofread and improved carefully, including all figures.

  1. The manuscript needs a revision in order to correct typos.

Answer: We appreciate that the reviewer pointed this out. The manuscript has been proofread and improved carefully.

Reviewer 3 Report

The manuscript describes in detail a computer-vision procedure based on deep neural networks for detection and tracking of ships, using the chimneys as reference points. The proposed system combines input images on different ranges (visible light and infrared) and applies state-of-the-art techniques for object detection. The results are good and improve other similar proposals.

While the paper is clearly organized and technically sound, there are a few issues to address in order to improve its content

1) Many acronyms (e.g., RPN, IOU, GIOU) are introduced without definition. Please, correct.

2) The KM association algorithm (Section 2.4.2) is too much briefly and poorly explained. It is difficult to follow the text and understand how it works. If possible, rewrite this part.

3) The computational load is not mentioned or discussed in the paper. It is, however, an important aspect of the proposed algorithm.

4) The bibliographics references are correct and up-to-date, but show a bias toward Chinese authors. The field is vast in literature.

Author Response

  1. Many acronyms (e.g., RPN, IOU, GIOU) are introduced without definition. Please, correct.

Answer: We appreciate that the reviewer pointed this out. The manuscript has been proofread and improved carefully. All acronyms (e.g., RPN, IOU, GIOU) are introduced with definitions. These revisions in the manuscript are highlighted in blue color.

  1. The KM association algorithm (Section 2.4.2) is too much brief and poorly explained. It is difficult to follow the text and understand how it works. If possible, rewrite this part.

Answer: We appreciate that the reviewer pointed this out. The KM association algorithm (Section 2.4.2) has been rewritten.

  1. The computational load is not mentioned or discussed in the paper. It is, however, an important aspect of the proposed algorithm.

Answer: We appreciate that the reviewer pointed this out. We have added the computational load in Section 4.4 and detailed the memory and CPU of the algorithm.

  1. The bibliographic references are correct and up-to-date, but show a bias toward Chinese authors. The field is vast in literature.

Answer: This is a very good suggestion. We have no prejudice against Chinese authors and thank Chinese authors for their outstanding contributions in this field. We have added a discussion of Chinese literature to our relevant work and referred to a large number of Chinese literature.

Round 2

Reviewer 1 Report

Thank you for your hard work.

This manuscript is a resubmission of an earlier submission. The following is a list of the peer review reports and author responses from that submission.